

# Physical development of infants born to patients with COVID-19 during pregnancy: 2 years of age

Anna Eligulashvili[1],[*], Moshe Gordon[1],[*], Sheri Nemerofsky[2], Tomas Havranek[2], Peter Bernstein[3], Judy Yee[1], Wei Hou[1] and Tim Duong[1]

[1] Radiology, Albert Einstein College of Medicine, Morris Park, NY, United States
[2] Department of Pediatrics, Division of Neonatology, Albert Einstein College of Medicine and the Children's Hospital at Montefiore, Bronx, New York, United States
[3] Department of Obstetrics, Gynecology, and Reproductive Science, Icahn School of Medicine at Mount Sinai, New York, New York, United States
[*] These authors contributed equally to this work.

## ABSTRACT

**Background:** SARS-CoV-2 infection during pregnancy and pandemic circumstances could negatively impact infant development. This study aimed to investigate the physical development, common pediatric illness incidence, and healthcare utilization over the first 2 years of life of infants born to COVID+ and COVID- patients. Comparisons were also made with infants born pre-pandemic.

**Methods:** This is a retrospective observational study at a major academic health system in New York City. Participants include all infants born to birthing persons with SARS-CoV-2 infection during pregnancy ($N = 758$) and without ($N = 9,345$) from 03/01/2020 to 08/17/2022, infants born pre-pandemic ($N = 3,221$) from 03/01/2017 to 08/17/2019, and birthing persons of all infants.

**Results:** There were no differences in weight, length, or head circumference curves between pandemic infants born to COVID+ and COVID- patients over the first 2 years of life ($p > 0.05$, repeated ANOVA). Annualized incidence of illness occurrence and healthcare utilization were similar between groups. Compared to pre-pandemic infants, the length of pandemic (COVID-) infants was lower from birth to 9 months ($p < 0.0001$). Pandemic infants additionally had more adverse perinatal outcomes including increased stillbirth (0.75% *vs.* 0.12%, $p = 0.0001$) and decreased gestational age (38.41 ± 2.71 *vs.* 38.68 ± 2.55 weeks, Cohen's d = −0.10, $p < 0.0001$), birthweight (2,597 ± 335 *vs.* 3,142 ± 643 g, Cohen's d = −1.06, $p < 0.0001$), and birth length (48.08 ± 4.61 *vs.* 49.09 ± 3.93 cm, Cohen's d = −0.24, $p < 0.0001$).

**Conclusions:** Birthing persons' SARS-CoV-2 infection status, birthing persons' profiles, and pandemic circumstances negatively affected perinatal outcomes, newborn physical development, and healthcare utilization. These findings draw clinical attention to the need to follow infants closely and implement enrichment to ensure optimal developmental outcomes.

Corresponding author
Tim Duong,
tim.duong@einsteinmed.edu

## INTRODUCTION

The COVID-19 pandemic raised concerns about the impact of the SARS-CoV-2 virus on pregnant patients and infants (*Allotey et al., 2020*; *Di Toro et al., 2021*; *Jafari et al., 2021*; *Mirbeyk, Saghazadeh & Rezaei, 2021*). Pregnant individuals with COVID-19 are at higher risk for severe illness and adverse outcomes (*Allotey et al., 2020*; *Metz et al., 2022*; *Nana & Nelson-Piercy, 2021*; *Rasmussen & Jamieson, 2022*). Potential overreactive host-mediated immune responses, cardiovascular and respiratory distress, and psychological stress from pandemic circumstances, could negatively affect fetal development *in utero* (*Gurol-Urganci et al., 2021*; *Papageorghiou et al., 2021*; *Seaton et al., 2023*; *Villar et al., 2021*; *Wei et al., 2021*). A few studies, including data from up to 18 different countries, reported SARS-CoV-2 infection during pregnancy was associated with increased preterm birth, preeclampsia, stillbirth, and low birth weight (*Gurol-Urganci et al., 2021*; *Papageorghiou et al., 2021*; *Seaton et al., 2023*; *Villar et al., 2021*; *Wei et al., 2021*). In addition, the pandemic could alter lifestyle, healthcare access, and social interaction that may affect fetal development *in utero* and infant development after birth. Some studies suggested that COVID-19 in pregnancy increases the risk of developmental and behavioral disorders or delays in children (*Edlow et al., 2023*, *2022*; *Fajardo Martinez et al., 2023*; *Hessami et al., 2022*; *Shah et al., 2023*; *Shuffrey et al., 2022*; *Zeng et al., 2021*). Data on long-term effects of COVID-19 disease during pregnancy on child development, health outcomes, and healthcare utilization are limited.

The long-term developmental impact of COVID-19 on infants remains unknown. The goal of this study was to investigate the physical development, incidence of common pediatric illnesses, and healthcare utilization in the first 2 years of life of infants born to patients with and without SARS-CoV-2 infection during pregnancy. The primary outcomes were physical development of infants measured by weight, length, and head circumference (HC). Secondary outcomes were annualized healthcare utilization (emergency department (ED), hospital admission, and outpatient visits), annualized common pediatric diagnoses and symptoms, and annualized infection incidence of infants. To assess whether the birthing persons' profiles affected outcomes, demographics, socioeconomic and insurance status, and prenatal care (PNC) adequacy were analyzed. A comparison was made to infants born prior to the COVID-19 pandemic to assess whether pandemic circumstances affected outcomes.

## MATERIALS AND METHODS

### Ethics

This retrospective observational study was approved by the Einstein-Montefiore Institutional Review Board (#2021-13658) with a waiver of informed consent. The data was accessed for current research purposes from October 14, 2022 to March 28, 2023 from electronic medical records (EMR) up to January 10, 2023. All authors had access to deidentified EMR data.

## Participants

Patients who delivered infants within the Montefiore Health System. Five groups were investigated: (i) infants of patients who were hospitalized for SARS-CoV-2 infection during gestation (COVID-hospitalized, $N = 39$), (ii) infants of patients who were infected with COVID-19, but not hospitalized due to infection during gestation (COVID-non-hospitalized, $N = 719$), (iii) all infants of patients who tested positive for COVID-19 during gestation, (COVID+, $N = 758$), (iv) infants of patients who tested negative for COVID-19 during gestation, (COVID-, $N = 9{,}345$), and (v) infants born prior to the pandemic (pre-pandemic, $N = 3{,}221$). For group i-iv, infants were born from 3/1/2020 to 8/17/2022 and for group v, infants were born from 3/1/2017 to 8/17/2019 (Fig. S1). SARS-CoV-2 infection was identified with real-time reverse transcriptase PCR-positive assay testing for SARS-CoV-2 RNA.

## Data sources

Health data came from the Montefiore Health System which included the Bronx and lower Westchester County (~10 miles diameter), serving a diverse population with the majority publicly insured. EMR were extracted (*Hoogenboom et al., 2021a*, *2021b*; *Iosifescu et al., 2022*; *Lu et al., 2023a*; *Lu, Hou & Duong, 2022*; *Lu et al., 2022*) and de-identified health data were obtained for research after standardization to the Observational Medical Outcomes Partnership (OMOP) Common Data Model (CDM) version 6.

## Variables

The profiles of pregnant individuals were tabulated. Demographics included age, body mass index (BMI), combined race and ethnicity, median household income quintile, and insurance status. Income data was obtained by matching patients' zip code to median household income as reported by the census, and then assigning quintiles based on the cohorts' combined incomes (*US Census Bureau, 2022*). Preexisting comorbidities included asthma, chronic obstructive pulmonary disease (COPD), congestive heart failure (CHF), chronic kidney disease (CKD), Types 1 and Type 2 diabetes mellitus, and hypertension that were designated by ICD-10 codes at delivery admission or prior. Adequacy of PNC was defined by the Kotelchuk Index (*Kotelchuck, 1994*).

The outcomes of infants were evaluated from birth till their age as of Jan 10, 2023. Birth measurements included gestational age (GA), hospitalization length of stay (LOS), weight, length, and HC. Birth outcomes included intensive care unit admission, invasive mechanical ventilation, preterm birth (GA <37 weeks), low birth weight (<2,500 g), stillbirth, and cesarean delivery. Outcomes after discharge from the birth hospitalization were tabulated and annualized to account for children of various ages. Follow-up outcomes included ED visits, hospitalization and average LOS, and outpatient visits within our healthcare system. Illness diagnoses included gastrointestinal (GI) symptoms (nausea, vomiting, and/or diarrhea), fever, jaundice, dehydration, bronchiolitis, and respiratory symptoms (nasal congestion, runny nose, sneezing, coughing, wheezing). Viral infection

by COVID-19, influenza, respiratory syncytial virus (RSV), pneumovirus, adenovirus, and methicillin-resistant *Staphylococcus aureus* (MRSA) was also tabulated.

All recorded measurements of weight, length, and HC were collected from the EMR and used for growth chart construction. All additional data was extracted from EMR.

## Analysis

Weight, length, and HC were measured as a function of time up to first 2 years of life. Measurements at 0 years were plotted as the mean average of all measurements taken from 0 to 1 month of age. After 1 month of age, measurements were averaged across 3-month bins (*i.e.*, 1–3 months) and plotted at the medians of timeframes. The sample size of growth chart measurements for each cohort at each timepoint is reported in Table S1. Cohorts included in the growth chart analysis were COVID-hospitalized, COVID-non-hospitalized, COVID+, COVID-, and separately, pandemic (COVID-) *vs.* pre-pandemic. Because the majority of patients delivered full-term newborns and very few were pre-term, there was no adjustment for age in growth chart analysis and measurements at birth (weight, length, height).

Birthing persons' data was compared between (i) COVID-hospitalized and COVID-non-hospitalized cohorts, (ii) COVID+ and COVID- cohorts, and (iii) pandemic (COVID-) and pre-pandemic cohorts. Infant data was compared amongst COVID+ and COVID- cohorts, and pandemic (COVID-) and pre-pandemic cohorts.

## Definition of COVID-19 waves

Predominant SARS-CoV-2 waves were estimated based on New York State Department of Health. Waves were defined by daily test positivity 5% above baseline and lasted at least 10 days in Bronx, New York. The first wave spanned from March 8, 2020, to May 25, 2020, the Alpha wave from December 6, 2020 to April 5, 2021, the Delta wave from July 6, 2021 to December 14, 2021, and the Omicron wave from December 15, 2021 to January 24, 2022 (*Lu et al., 2023b*; *Seaton et al., 2023*).

## Outcomes

Primary outcomes were weight, length, and HC from birth to 2 years of age. Secondary outcomes included annualized healthcare utilization (ED, hospital admission, and outpatient visits) and incidences of common pediatric illnesses and symptoms across different COVID-19 as detailed above.

Tertiary outcomes included the effects of birthing people demographics, socioeconomic status (SES), insurance status, PNC, and the effects of pandemic circumstances.

## Statistical analysis

Statistical analysis was performed using RStudio. Group comparison for categorical variables used $\chi^2$ or Fisher's exact tests, and group comparison for continuous variables used Wilcoxon test. Repeated measures ANOVA was performed for growth chart comparison with appropriate *post hoc* analysis. $P < 0.05$ were statistically significant.

## Data sharing

Deidentified individual participant data will be made available upon publication to researchers who provide a methodologically sound proposal for use in achieving the goals of the approved proposal. Proposals should be submitted to the corresponding author.

## RESULTS

### Infants during the pandemic

Figure 1 displays physical development as measured by weight, length, and HC of infants over their first 2 years of life for COVID+, COVID-, COVID-hospitalized, and COVID-non-hospitalized cohorts. There were no group differences in growth charts between any pairs ($p > 0.05$, repeated ANOVA).

Table 1 shows the secondary outcomes of infants born to COVID+ and COVID-mothers. There were no differences in measurements and outcomes at birth, except that the incidence of stillbirth was lower in COVID+ patients (0% *vs.* 0.87%, $p = 0.017$). There were no differences in healthcare utilization ($p > 0.05$). Infants of COVID+ patients had lower annualized bronchiolitis (0.17 ± 0.82 *vs.* 0.26 ± 1.04, $p = 0.002$) and pneumovirus (0 ± 0 *vs.* 0.002 ± 0.045, $p < 0.0001$), but no other differences in symptom and disease incidence or viral infection. Table S2 shows these secondary outcomes broken down by wave of infection and similarly revealed no major differences in any variables.

### Birthing persons' profiles

To evaluate whether birthing persons' profiles affected outcomes, demographics, SES, insurance status, and PNC were analyzed (Table 2). The only difference between the COVID+ and COVID- was the lower proportions of White, non-Hispanic patients (1.98% *vs.* 3.98%, $p = 0.008$) in the COVID+. The COVID+ had overall similar household median income distributions but lower proportion of patients within the 5th quintile as compared to COVID- patients (13.72% *vs.* 18.44%, $p = 0.001$). COVID+ patients had higher incidence of asthma (20.45% *vs.* 14.47%, $p < 0.0001$) and lower incidence of inadequate PNC (37.60% *vs.* 41.01%, $p = 0.020$).

There were no statistically significant differences between COVID-hospitalized and COVID-non-hospitalized patients.

### Comparison with pre-pandemic infants

To assess whether pandemic circumstances affected outcomes, comparisons were made with infants born pre-pandemic. The physical development measured by length ($p < 0.0001$) differed, but not weight or HC over 2 years (Fig. 2); infants born during the pandemic (COVID-) were shorter than pre-pandemic infants from birth to 9 months ($p < 0.0001$ at birth, $p = 0.0005$ at 1–3 months, $p = 0.016$ at 4–6 months, and $p = 0.046$ at 7–9 months).

Table 3A presents the secondary outcomes between pandemic (COVID-) and pre-pandemic infants. Pandemic infants had lower GA (38.41 ± 2.71 *vs.* 38.68 ± 2.55 weeks, Cohen's d = −0.10, $p < 0.0001$), LOS (2.33 ± 1.85 *vs.* 2.42 ± 2.24 days, $p = 0.020$), birthweight (2,597.50 ± 335.88 *vs.* 3,142.03 ± 643.92 g, Cohen's d = −1.06, $p < 0.0001$), and

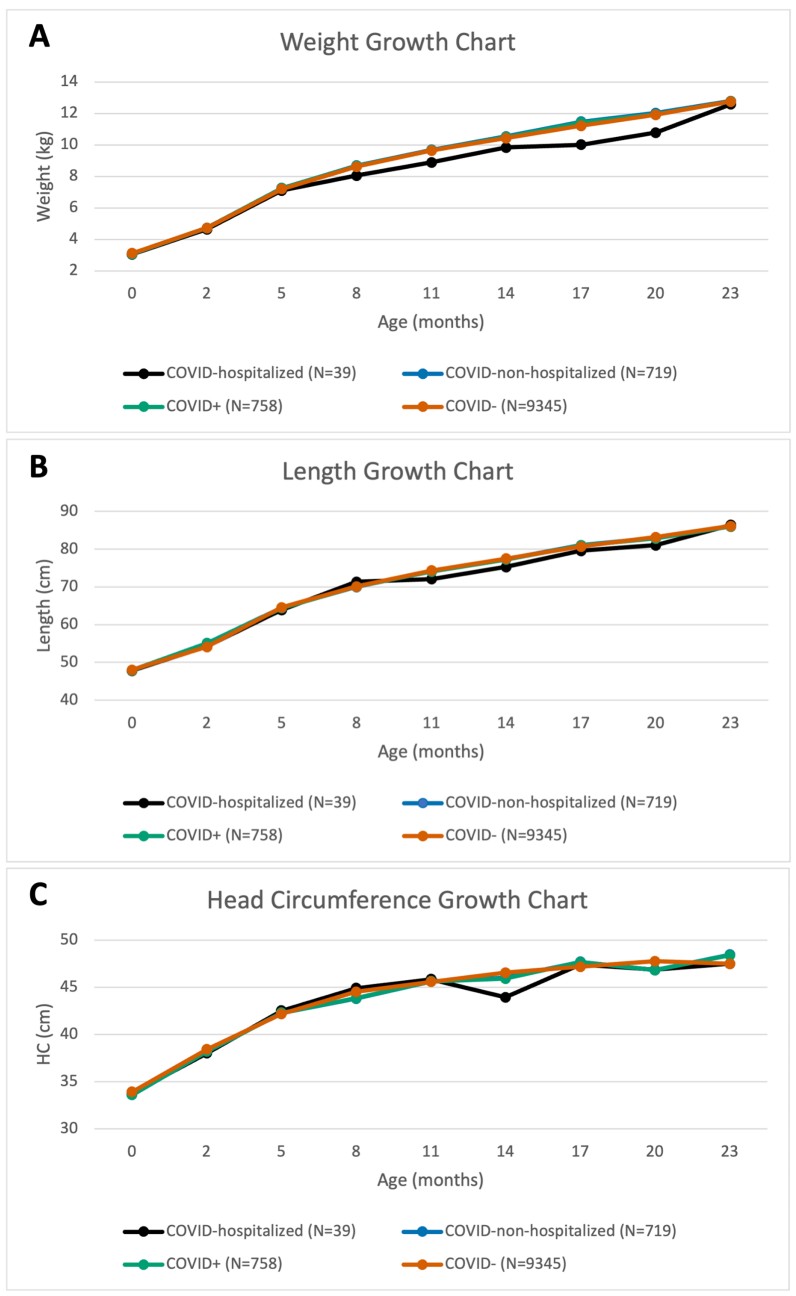

**Figure 1 Pandemic cohort.** Growth charts of (A) weight, (B) length, and (C) head circumference for various pandemic cohorts over the first 24 months of life.

birth length (48.08 ± 4.61 *vs.* 49.09 ± 3.93 cm, Cohen's d = −0.24, $p < 0.0001$). Pandemic infants had increased incidence of stillbirth (0.75% *vs.* 0.12%, $p = 0.0001$) and were delivered by cesarean more frequently (35.88% *vs.* 32.47%, $p = 0.0005$).

At follow-up, pandemic infants had fewer ED visits (1.59 ± 2.80 *vs.* 2.52 ± 2.20, $p < 0.0001$) and increased outpatient visits (16.94 ± 17.87 *vs.* 15.13 ± 14.54, 0 = 0.002). With respect to illnesses, pandemic infants experienced fewer GI symptoms (0.29 ± 0.93 *vs.* 0.35 ± 1.23, $p = 0.006$), fever (0.23 ± 0.74 *vs.* 0.35 ± 0.92, $p < 0.0001$), dehydration (0.03 ±

**Table 1 Infant outcomes between COVID+ and COVID- groups.**

| | COVID+ (n = 758) | COVID- (n = 9,345) |
|---|---|---|
| Measurements at birth | | |
| Gestational age (weeks) | 38.28 ± 2.54 | 38.41 ± 2.71 |
| LOS (days) | 2.51 ± 2.42 | 2.33 ± 1.85 |
| Weight (g) | 3,093.06 ± 701.78 | 2,597.5 ± 335.88 |
| Length (cm) | 47.93 ± 4.20 | 48.08 ± 4.61 |
| Head circumference (cm) | 33.60 ± 2.29 | 33.91 ± 16.79 |
| Outcomes at birth | | |
| Critical care | 40 (5.28%) | 393 (4.21%) |
| Preterm birth | 109 (14.38%) | 1,211 (12.96%) |
| Birth weight <2,500 g | 106 (14%) | 1,145 (12.25%) |
| Stillbirth | 0 (0%) | 70 (0.75%)* |
| Cesarean delivery | 298 (39.31%) | 3,354 (35.88%) |
| Annualized outcomes | | |
| ED visit | 2.03 ± 3.10 (394) | 1.59 ± 2.80 (4,227) |
| Hospitalization | 1.59 ± 1.43 (758) | 1.32 ± 1.35 (9,256) |
| LOS (days) | 3.22 ± 7.62 | 3.20 ± 7.90 |
| Outpatient visits | 19.09 ± 16.06 (754) | 16.94 ± 17.87 (9,233) |
| Symptom/disease | | |
| GI symptoms | 0.26 ± 1.02 (104) | 0.29 ± 0.93 (1,185) |
| Fever | 0.25 ± 0.71 (123) | 0.23 ± 0.74 (1,324) |
| Jaundice | 0.60 ± 2.19 (154) | 0.53 ± 1.69 (1,700) |
| Dehydration | 0.02 ± 0.22 (16) | 0.03 ± 0.21 (141) |
| Bronchiolitis | 0.17 ± 0.82 (72) | 0.26 ± 1.04** (704) |
| Respiratory symptoms | 0.63 ± 1.45 (197) | 0.53 ± 1.44 (2,139) |
| Viral infection | | |
| SARS-CoV-2 | 0.07 ± 0.25 (65) | 0.05 ± 0.21 (696) |
| Influenza | 0.15 ± 0.51 (100) | 0.11 ± 0.44 (1,020) |
| RSV | 0.08 ± 0.51 (1,020) | 0.05 ± 0.38 (273) |
| Pneumovirus | 0 ± 0 (25) | 0.0023 ± 0.045*** (28) |
| Adenovirus | 0.0037 ± 0.06 (4) | 0.004 ± 0.05 (51) |
| MRSA | 0.0026 ± 0.045 (3) | 0.006 ± 0.13 (37) |

Note:
$*p < 0.05$, $**p < 0.01$, $***p < 0.001$ between infants born to COVID+ vs. COVID- patients. Data reported as mean ± standard deviation or N (%). Annualized outcomes also report number of patients with outcome of interest (N).

**Table 2 Birthing persons' profiles.**

| | COVID-negative (N = 9,345) | COVID-positive (N = 758) | COVID-hospitalized (N = 39) | COVID-non-hospitalized (N = 719) |
|---|---|---|---|---|
| Demographics | | | | |
| Age | 31.94 ± 6.05 | 31.86 ± 9.58 | 33.90 ± 5.15 | 31.75 ± 6.00 |
| BMI | 31.02 ± 6.46 | 31.59 ± 6.67 | 33.54 ± 8.20 | 31.47 ± 6.57 |

(Continued)

| | COVID-negative (N = 9,345) | COVID-positive (N = 758) | COVID-hospitalized (N = 39) | COVID-non-hospitalized (N = 719) |
|---|---|---|---|---|
| Combined race and ethnicity | | | | |
| Hispanic | 4,504 (48.20%) | 398 (52.51%) | 15 (38.46%) | 383 (53.27%) |
| Black, non-Hispanic | 2,535 (27.13%) | 230 (30.34%) | 15 (38.46%) | 215 (29.90%) |
| White, non-Hispanic | 372 (3.98%)^^ | 15 (1.98%)^^ | 0 (0%) | 15 (2.09%) |
| Other | 1,958 (20.95%)^^^ | 117 (15.44%)^^^ | 9 (23.08%) | 108 (15.02%) |
| Comorbidities | | | | |
| Asthma | 1,352 (14.47%)^^^ | 155 (20.45%)^^^ | 10 (25.64%) | 145 (20.17%) |
| COPD | 11 (0.12%) | 2 (0.26%) | 0 (0%) | 2 (0.28%) |
| Chronic heart failure | 18 (0.19%) | 4 (0.52%) | 0 (0%) | 4 (0.56%) |
| Chronic kidney disease | 34 (0.36%) | 4 (0.52%) | 0 (0%) | 4 (0.56%) |
| Diabetes | 1,483 (15.87%) | 118 (15.57%) | 10 (25.64%) | 108 (15.02%) |
| Hypertension | 732 (7.83%) | 64 (8.44%) | 2 (5.13%) | 62 (8.62%) |
| Median household income quintile (upper limit) | | | | |
| 1 ($ 34,860) | 2,030 (21.72%) | 169 (22.30%) | 6 (15.38%) | 163 (22.67%) |
| 2 ($ 40,138) | 2,077 (22.23%) | 167 (22.03%) | 13 (33.33%) | 154 (21.42%) |
| 3 ($ 55,553) | 1,536 (16.44%) | 131 (17.28%) | 3 (7.69%) | 128 (7.80%) |
| 4 ($ 62,918) | 1,973 (21.11%) | 181 (23.88%) | 9 (23.08%) | 172 (23.92%) |
| 5 ($ 250,001) | 1,723 (18.44%)^^ | 104 (13.72%)^^ | 8 (20.51%) | 96 (13.35%) |
| Insurance status | | | | |
| Private insurance | 1,607 (17.20%) | 131 (17.28%) | 8 (20.51%) | 123 (17.11%) |
| Medicaid | 6,993 (74.83%) | 573 (75.59%) | 29 (74.36%) | 544 (75.66%) |
| Medicare | 46 (0.49%) | 1 (0.13%) | 0 (0%) | 1 (0.14%) |
| Care Management Organization | 355 (3.80%) | 32 (4.22%) | 1 (2.56%) | 31 (4.31%) |
| Uninsured | 215 (2.30%) | 18 (2.37%) | 1 (2.56%) | 17 (2.36%) |
| Other | 4 (0.04%) | 0 (0%) | 0 (0%) | 0 (0%) |
| Kotelchuk PNC Index | | | | |
| Adequate plus | 3,970 (42.48%) | 347 (45.78%) | 21 (53.85%) | 326 (45.34%) |
| Adequate | 951 (10.18%) | 91 (12.01%) | 4 (10.26%) | 87 (12.10%) |
| Intermediate | 493 (5.28%) | 35 (4.62%) | 1 (2.56%) | 34 (4.73%) |
| Inadequate | 3,926 (42.01%)^ | 285 (37.60%)^ | 13 (33.33%) | 272 (37.83%) |

**Note:**

^$p < 0.05$, ^^$p < 0.01$, ^^^$p < 0.001$ between COVID-positive and COVID-negative. Data reported as mean ± standard deviation or N (%).

0.21 *vs.* 0.04 ± 0.16, $p = 0.003$), and respiratory symptoms (0.53 ± 1.44 *vs.* 0.16 ± 0.34, $p = 0.002$). The only viral infection difference seen was in the pandemic cohort with a lower incidence of influenza (0.11 ± 0.44 *vs.* 0.16 ± 0.34, $p < 0.0001$).

## Comparison with pre-pandemic birthing persons' profiles

The birthing persons' profiles were assessed between COVID- and pre-pandemic patients (Table 3B). The pandemic cohort had higher proportions of median household incomes in the 1st quintile (21.72% *vs.* 17.88%, $p < 0.0001$), lower proportions in the 3rd quintile

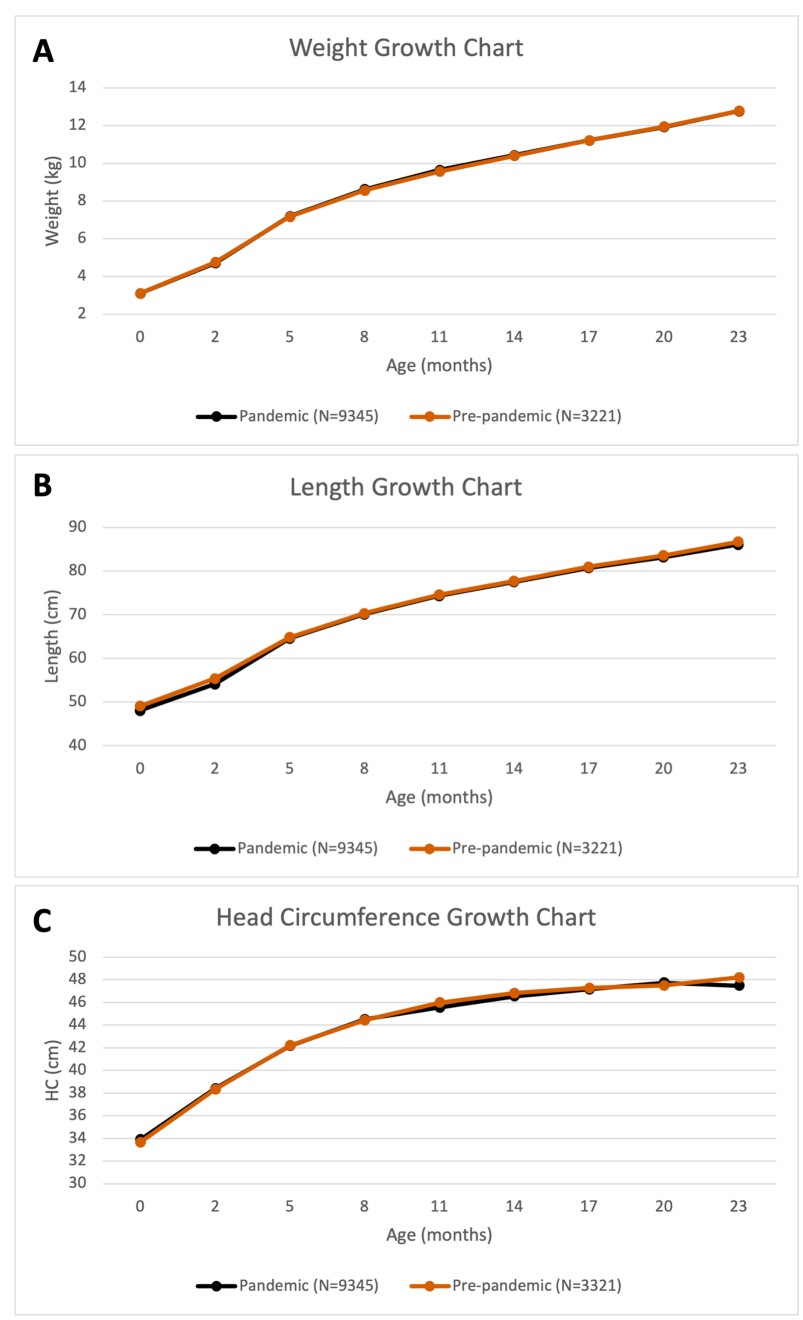

**Figure 2 Pre-pandemic comparison.** Growth charts of (A) weight, (B) length, and (C) head circumference for pandemic (infants born to COVID- patients) and pre-pandemic cohorts over the first 24 months of life.

(16.44% *vs.* 18.97%, *p* = 0.009), lower Medicaid use (74.83% *vs.* 77.46%, *p* = 0.003) and higher uninsured status (2.30% *vs.* 1.55%, *p* = 0.013). The pandemic cohort showed lower adequate PNC (10.18% *vs.* 14.16%, *p* < 0.0001) and higher inadequate PNC (42.01% *vs.* 34.65%, *p* < 0.0001).

**Table 3 Pre-pandemic comparison.**

| (A) | Pandemic (COVID-) (*N* = 9,345) | Pre-pandemic (*N* = 3,221) |
|---|---|---|
| Measurements at birth | | |
| Gestational age (weeks) | 38.41 ± 2.71*** | 38.68 ± 2.55*** |
| LOS (days) | 2.33 ± 1.85* | 2.42 ± 2.24* |
| Weight (g) | 2,597.5 ± 335.88*** | 3,142.03 ± 643.92*** |
| Length (cm) | 48.08 ± 4.61*** | 49.09 ± 3.93*** |
| HC (cm) | 33.91 ± 16.79 | 33.67 ± 6.57 |
| Outcomes at birth | | |
| Critical care | 393 (4.21%) | 118 (3.66%) |
| Preterm birth | 1,211 (12.96%) | 381 (11.83%) |
| Birth weight <2,500 g | 1,145 (12.25%) | 371 (11.52%) |
| Stillbirth | 70 (0.75%) | 4 (0.12%)*** |
| Cesarean delivery | 3,354 (35.88%)*** | 1,046 (32.47%)*** |
| Annualized outcomes | | |
| ED Visit | 1.59 ± 2.80*** (4,227) | 2.52 ± 2.20*** (3,120) |
| Hospitalization | 1.32 ± 1.35 (9,256) | 1.28 ± 0.92 (205) |
| LOS (days) | 3.20 ± 7.90 | 1.36 ± 2.79 |
| Outpatient visits | 16.94 ± 17.87** (9,223) | 15.13 ± 14.54** (3,169) |
| Symptom/disease | | |
| GI Symptoms | 0.29 ± 0.93** (1,185) | 0.35 ± 1.23** (1,119) |
| Fever | 0.23 ± 0.74*** (1,324) | 0.35 ± 0.92*** (1,360) |
| Jaundice | 0.53 ± 1.69 (1,700) | 0.15 ± 0.47 (533) |
| Dehydration | 0.03 ± 0.21** (141) | 0.04 ± 0.16** (209) |
| Bronchiolitis | 0.26 ± 1.04 (704) | 0.19 ± 0.50 (731) |
| Respiratory symptoms | 0.53 ± 1.44** (2,139) | 0.60 ± 1.08** (1,610) |
| Viral infection | | |
| COVID-19 | 0.05 ± 0.21 (696) | 0.02 ± 0.08 (247) |
| Influenza | 0.11 ± 0.44*** (1,020) | 0.16 ± 0.34*** (928) |
| RSV | 0.05 ± 0.38 (273) | 0.05 ± 0.20 (260) |
| Pneumovirus | 0.002 ± 0.045 (28) | 0.0007 ± 0.0154 (8) |
| Adenovirus | 0.004 ± 0.05 (51) | 0.002 ± 0.02 (27) |
| MRSA | 0.006 ± 0.13 (37) | 0.0051 ± 0.097 (22) |
| **(B)** | **Pandemic (COVID-) (*N* = 9,345)** | **Pre-pandemic (*N* = 3,221)** |
| Demographics | | |
| Age | 31.94 ± 6.05*** | 33.79 ± 6.08*** |
| BMI | 31.02 ± 6.46 | 31.05 ± 6.46 |
| Combined Race and Ethnicity | | |
| Hispanic | 4,504 (48.20%) | 1,645 (51.07%) |
| Black, non-Hispanic | 2,535 (27.13%)* | 969 (30.08%)* |
| White, non-Hispanic | 372 (3.98%)*** | 70 (2.17%)*** |

| (B) | Pandemic (COVID-) (N = 9,345) | Pre-pandemic (N = 3,221) |
|---|---|---|
| Other | 1,958 (20.95%)*** | 546 (16.95%)*** |
| Comorbidities | | |
| Asthma | 1,352 (14.47%)*** | 582 (18.07%)*** |
| COPD | 11 (0.12%)*** | 15 (0.47%)*** |
| Chronic heart failure | 18 (0.19%)* | 15 (0.47%)* |
| Chronic kidney disease | 34 (0.36%)** | 27 (0.84%)** |
| Diabetes | 1,483 (15.87%)** | 453 (14.06%)** |
| Hypertension | 732 (7.83%)** | 320 (7.30%)** |
| Median household income quintile (upper limit) | | |
| 1 ($ 34,860) | 2,030 (21.72%)*** | 576 (17.88%)*** |
| 2 ($ 40,138) | 2,077 (22.23%) | 733 (22.76%) |
| 3 ($ 55,553) | 1,536 (16.44%)** | 611 (18.97%)** |
| 4 ($ 62,918) | 1,973 (21.11%) | 737 (22.88%) |
| 5 ($ 250,001) | 1,723 (18.44%) | 562 (17.45%) |
| Insurance status | | |
| Private insurance | 1,607 (17.20%) | 505 (15.68%) |
| Medicaid | 6,993 (74.83%)** | 2,495 (77.46%)** |
| Medicare | 46 (0.49%) | 25 (0.78%) |
| Care Management Organization | 355 (3.80%) | 143 (4.44%) |
| Uninsured | 215 (2.30%)$ | 50 (1.55%)* |
| Other | 4 (0.04%) | 0 (0%) |
| Kotelchuk PNC index | | |
| Adequate plus | 3,970 (42.48%) | 1,471 (45.67%) |
| Adequate | 951 (10.18%)*** | 456 (14.16%)*** |
| Intermediate | 493 (5.28%) | 174 (5.40%) |
| Inadequate | 3,926 (42.01%)*** | 1,116 (34.65%)*** |

Note:
(A) Infant outcomes by birth during (3/1/2020–8/17/2022) or before (3/1/2017–8/17/2019) the pandemic. Annualized outcomes also report number of patients with outcome of interest (N). (B) [b]Birthing persons' profiles by delivery date during pandemic (COVID-) and pre-pandemic. *$p < 0.05$, **$p < 0.01$, ***$p < 0.001$ between pandemic (COVID-) and pre-pandemic. Data reported as mean ± standard deviation or N (%).

## DISCUSSION

This study investigated the physical development, healthcare utilization, and illness incidence of infants born to COVID+ individuals until 2 years of age. Effects of pregnant individuals' profiles and pandemic circumstances on infant outcomes were evaluated. There were no significant differences in physical development between infants born to COVID+ and COVID- patients. Healthcare utilization and illness incidence over the first 2 years of life were generally not different. Patient profiles influenced perinatal and infant outcomes. Pandemic circumstances negatively affected perinatal outcomes, physical development, and healthcare utilization.

## COVID-19 exposure during pregnancy could negatively affect development *in utero*

SARS-CoV-2 RNA has been reported in stool of healthy infants born to patients with COVID-19 during pregnancy, pointing to transmission of the virus *in utero* and presence of infection in newborn intestines (*Jin et al., 2022*). Most infants born to infected patients however tested negative (*Thomas et al., 2022*; *Vivanti et al., 2020*). Nonetheless, maternal SARS-CoV-2 infection was reported to be associated with placental abnormalities including fetal vascular malperfusion (*Patberg et al., 2021*). SARS-CoV-2 infection during pregnancy may be associated with preeclampsia, which can lead to adverse maternal and fetal outcomes (*Papageorghiou et al., 2021*; *Rana et al., 2019*). COVID-19 in pregnancy may also be associated with substantial increases in severe maternal morbidity and mortality (*Villar et al., 2021*). A meta-analysis study reported an association between SARS-CoV-2 infection and preterm birth, preeclampsia, stillbirth, and low birth weight, indicating that infection can impact perinatal outcomes (*Wei et al., 2021*). In addition, COVID-19 related hyperactive host-mediated immune response, hypoxia, cardiovascular and respiratory distress experienced during pregnancy could negatively affect *in utero* development. A hyperinflammatory systemic response to SARS-CoV-2 infection may make the uterus less hospitable for fetal growth and development. Moreover, psychological distress associated with COVID-19 disease during pregnancy could negatively affect development *in utero*, including disruptions in homeostasis that lead to increased probability of preterm birth and preeclampsia (*Traylor et al., 2020*). Our findings on perinatal outcomes agree with the literature. The surge in premature birth may be explained by iatrogenic preterm deliveries to improve birthing persons' respiratory status in cases of COVID-19 illness.

This study is the longest follow-up of physical development in infants. A previous study found no physical growth abnormalities of neonates born to COVID+ patients up to 44 weeks GA and without comparison to neonates born to COVID- patients (*Zeng et al., 2021*). Although it is reassuring that infants born to patients with SARS-CoV-2 infection do not show any immediate physical abnormalities, there is still concern for long-term effects of *in utero* exposure that may not be apparent. There were no significant differences in healthcare utilization, pediatric illnesses, or viral infection incidence between the COVID+ and COVID- groups. There are no similar studies with which to compare.

## Birthing persons' status could negatively affect infant development

Patient profiles, including demographics, comorbidities, SES, insurance status, and PNC could contribute to negative infant outcomes. COVID+ patients in our cohort had higher asthma incidence and lower SES, both of which are associated with pregnancy complications (*Bonham, Patterson & Strek, 2018*; *Silva et al., 2008*; *Wang, Li & Huang, 2020*). Very few patients were hospitalized due to COVID-19 and there were no associations in any variables with hospitalization status. Despite younger age and fewer comorbidities compared to the general adult population affected by COVID-19, COVID+ pregnant individuals may be more vulnerable to severe disease, contributing to differences

in newborn outcomes (*Hoogenboom et al., 2021a*, *2021b*; *Iosifescu et al., 2022*; *Lu et al., 2023a*; *Lu, Hou & Duong, 2022*; *Lu et al., 2022*).

## Socioeconomic status

Lower SES contributes to worse COVID-19 and pregnancy outcomes in the general population (*Acosta et al., 2021*; *Kanwal et al., 2022*; *Kim et al., 2018*; *Magesh et al., 2021*; *Sow, Raynault & De Spiegelaere, 2022*). The COVID+ patients had slightly lower incomes compared to COVID- patients, highlighting SES that may have partially contributed to differences with COVID- patients. Surprisingly, COVID- patients had higher inadequate PNC, suggesting that they might have felt healthy and avoided potential COVID-19 exposure at healthcare facilities.

Pandemic patients had lower SES compared to pre-pandemic patients, potentially contributing to decreased adequate PNC. The decrease in SES could be due in part to the outflux of affluent families from New York City, paired with an increase in unemployment incidence (*Schleimer et al., 2022*). The socioeconomic implications of the pandemic characterized by decreased access to healthcare in disadvantaged communities, such as the population in the Bronx (*Peng et al., 2024*; *Eligulashvili et al., 2024*; *Hadidchi et al., 2024*), are in agreement with the literature (*Bambra et al., 2020*; *Roberts & Tehrani, 2020*; *Song et al., 2024*). Studies have previously reported an increase in the number of uninsured people during the pandemic with respect to both the general population and specifically post-partum (*Bundorf, Gupta & Kim, 2021*; *Eliason, Daw & Steenland, 2022*). Our findings suggest that lower SES could be associated with developmental changes of infants born in the pandemic. Such decreases in SES could result in decreased healthcare utilization, which may explain the differences in PNC and infant follow-up outcomes. The lower adequacy of PNC might also be due to the decreased willingness of healthy patients to enter healthcare facilities to avoid being infected. It is not surprising that the level of adequate PNC during the pandemic was lower compared to pre-pandemic given the limited accessibility to healthcare and increased safety restrictions.

## Circumstances surrounding the pandemic could negatively affect infant development

The COVID-19 pandemic starkly impacted healthcare accessibility and societal norms, leading to the question of how this dramatic environmental change affected pregnancy and infant outcomes. Environmental stressors can increase anxiety in expectant patients, which may lead to an imbalance of homeostasis mediators and increase the probability of pregnancy complications (*Traylor et al., 2020*). A systematic review found that pregnancies during the pandemic experienced a reduction in preterm birth and increase in mean birthweight, but no increase in stillbirth (*Yang et al., 2022*).

There is currently no literature on how pandemic infant outcomes differ from pre-pandemic outcomes with respect to physical development, healthcare utilization, and illnesses incidence. The pandemic infants showed decreases in ED usage and incidences of many illnesses, including influenza. Concerns of SARS-CoV-2 infection at healthcare facilities could explain the decreased use of ED. Avoidance of social gatherings during the

pandemic could explain the marked reduction in influenza infection and other commutable illnesses. Taken together, pandemic circumstances *per se* affect the incidence of pediatric disorders and healthcare utilization, which may further contribute to the long-term development of infants.

### Limitations

This study has several limitations. Infants who did not return to our health system after birth could not be studied. Due to varying levels of PNC and COVID-19 testing availability, this study was more likely to detect cases at the end of pregnancy since universal testing was performed at delivery. Outcomes could be affected by vaccination, wave of infection, COVID-19 testing rate, and population profile, among others. Vaccination data was not available if performed outside the healthcare system and thus vaccination status was not analyzed with respect to outcomes. The reporting of several viral infections may be biased as testing may have required severe illness and/or hospitalization. Given the sample size, retrospective nature of the study, and the evolving treatment options across the pandemic, this study is underpowered to test the effects of different COVID-19 therapies. As with any retrospective study, there could be unintentional confounders and bias.

## CONCLUSIONS

This is one of the largest and longest follow-up studies of physical development, illness incidence, and healthcare utilization of infants born to COVID+ patients. Some primary and secondary outcomes were affected by birthing persons' COVID-19 status, patient profiles, and pandemic circumstances. While it is reassuring that infants born to SARS-CoV-2 infected patients showed normal physical development up to 2 years of age, longer term effects of *in utero* exposure to COVID-19 disease are still unknown. Longer follow-up studies, that include mental and cognitive development assessments are warranted.

### Funding

The authors received no funding for this work.

### Competing Interests

The authors declare that they have no competing interests.

### Author Contributions

- Anna Eligulashvili conceived and designed the experiments, performed the experiments, analyzed the data, prepared figures and/or tables, authored or reviewed drafts of the article, and approved the final draft.
- Moshe Gordon conceived and designed the experiments, performed the experiments, analyzed the data, prepared figures and/or tables, authored or reviewed drafts of the article, and approved the final draft.

- Sheri Nemerofsky conceived and designed the experiments, performed the experiments, analyzed the data, authored or reviewed drafts of the article, and approved the final draft.
- Tomas Havranek conceived and designed the experiments, authored or reviewed drafts of the article, and approved the final draft.
- Peter Bernstein conceived and designed the experiments, performed the experiments, analyzed the data, authored or reviewed drafts of the article, and approved the final draft.
- Judy Yee conceived and designed the experiments, prepared figures and/or tables, authored or reviewed drafts of the article, and approved the final draft.
- Wei Hou conceived and designed the experiments, performed the experiments, analyzed the data, authored or reviewed drafts of the article, and approved the final draft.
- Tim Duong conceived and designed the experiments, performed the experiments, analyzed the data, prepared figures and/or tables, authored or reviewed drafts of the article, and approved the final draft.

## Human Ethics

The following information was supplied relating to ethical approvals (*i.e.*, approving body and any reference numbers):

Einstein-Montefiore Institutional Review Board (#2021-13658).

## Data Availability

The raw data are available in the Supplemental Files.

## Supplemental Information

Supplemental information for this article can be found online at http://dx.doi.org/10.7717/peerj.18481#supplemental-information.

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
