# Peer review of "Physical development of infants born to patients with COVID-19 during pregnancy: 2 years of age"

_PeerJ, doi:10.7717/peerj.18481_

## Round 0.1 · original submission · Major Revisions

Dear Dr. Eligulashvili and colleagues:

Thanks for submitting your manuscript to PeerJ. I have now received three independent reviews of your work, and as you will see, the reviewers raised some minor concerns about the manuscript. Despite this, these reviewers are optimistic about your work and the potential impact it will have on research studying the impact of COVID-19 on newborns and children of infected patients. Thus, I encourage you to revise your manuscript, accordingly, considering all the concerns raised by all reviewers.

There are several important suggestions, which I am sure will greatly improve your manuscript once addressed. The overall problem seems to be a presentation lacking clarity, so please address areas where clarity is needed. There also seems to be a problem with your interpretation of statistical significance (per reviewer 2). Please also address the confounding factors raised by reviewer 3.

Please use the comments by the reviewers to add missing information where noted. Try to restructure your manuscript for clarity, avoiding redundancy and streamlining sections for effective delivery. Missing references should be added.

I look forward to seeing your revision, and thanks again for submitting your work to PeerJ.

Good luck with your revision,

-joe

Reviewer 1 ·

Basic reporting

This paper reported a large scale and retrospective study on the effect of birthing person’s COVID-19 infection on new borns. The results demonstrate that birthing person’s COVID-19 contraction does not have strong side effects on the new babies, which is assuring. Meanwhile, the pandemic itself affect all of the new borns in general. The study is very interesting and important on a timely manner. It is very novel and worthy of immediate publication for public attention.

Experimental design

This is a retrospective study. It is also the largest study I have ever read so far on this topic. The study was designed very well. Statistical analysis is also very reasonable and well thought.

Validity of the findings

The findings are very solid and important, based on a reasonable design and rigorous analysis.

Additional comments

Well done study. Important finding. Suggest promote it to public media due the nature of this topic.

Reviewer 2 ·

Basic reporting

.

Experimental design

.

Validity of the findings

.

Additional comments

Dear author,
1. Please specify the exact type of study. This appears to be a historical cohort study.
2. Explain the sampling method clearly, please.
3. How did you check the severity of Covid-19 in the affected group? Was the hospitalization of the neonates your only criterion? Hasn't blood oxygen saturation been used in neonates?
4. How did you check the normal distribution of the data for analysis?
5. In the Result section, "gestational age (38.41±2.71 vs 38.68±2.55 weeks, p<0.0001) "does not appear to be true.

Reviewer 3 ·

Basic reporting

In this manuscript, the authors investigated the physical development of infants born to parents with covid-19 during pregnancy and also compared those born before and during the pandemic with a retrospective observation stud using data from a major health system in NY. The manuscript is generally well written and the reporting is clear.

Experimental design

My main concern with this manuscript is the potential influence of confounding factors. As the authors show in the tables, there are multiple covariates being significantly different among the groups that are compared. As such, it is highly likely that these covariates may have been driving the main findings reported in this paper.

Validity of the findings

see above

Additional comments

1, The authors are encouraged to also report effect size measures in the abstract, in addition to p values.
2, introduction: the authors reviewed multiple studies of the influence of sars-cov-2 in the first paragraph, can you give more information regarding in what countries and the timing of the exposure?
3, methods and results: the period from 3/1/2020 to 8/17/2022 is quite broad and it is obvious that those born early during the pandemic will be affected to a greater degree. given that the sample size of the study is rather large, if possible, can you authors conduct addition analysis by further categorizing this period?
4, methods: regarding preexisting comorbidities reported in lines 95-98, is there information on psychiatric disorders available?

---

## Round 0.2 · accepted · Accept

Dear Dr. Eligulashvili and colleagues:

Thanks for revising your manuscript based on the concerns raised by the reviewers. I now believe that your manuscript is suitable for publication. Congratulations! I look forward to seeing this work in print, and I anticipate it being an important resource for groups studying the impact of COVID-19 on newborns and children of infected patients.. Thanks again for choosing PeerJ to publish such important work.

Best,

-joe

Reviewer 3 ·

Basic reporting

Thank the authors for addressing my concerns.

Experimental design

Good

Validity of the findings

Good